# Mechanism and Purification Effect of Photocatalytic Wastewater Treatment Using Graphene Oxide-Doped Titanium Dioxide Composite Nanomaterials

Sheng Liu [ID], Zi-Lin Su, Yi Liu, Lin-Ya Yi, Zhan-Li Chen * and Zhen-Zhong Liu

School of Resources Environment and Chemical Engineering, Nanchang University, 999 Xuefu Avenue, Nanchang 330031, China; 6003118044@email.ncu.edu.cn (S.L.); 6003118035@email.ncu.edu.cn (Z.-L.S.); ly1539845788@163.com (Y.L.); 6003118041@email.ncu.edu.cn (L.-Y.Y.); 6003118046@email.ncu.edu.cn (Z.-Z.L.)
* Correspondence: karlhigh@vip.163.com

**Abstract:** The present work aims to examine the mechanism and purification effect of graphene oxide (GO) and GO composite materials for photocatalysis sewage treatment. $TiO_2$ nanoparticles were prepared using the sol-gel technique; GO was prepared using the modified Hummers technique; and finally, a new $N-TiO_2/GO$ photocatalysis composite material was prepared by hydrothermal synthesis. As a nitrogen source, urea uses non-metal doping to broaden the photoresponse range of $TiO_2$. The prepared GO and its composite materials are characterized. Simulation experiments, using the typical water dye pollutant rhodamine B (RhB), tested and analyzed the adsorption and photocatalysis performances of the prepared GO and its composite materials. Characterization analysis demonstrates that $TiO_2$ is distributed on the GO surface in the prepared $N-TiO_2/GO$ material. Simultaneously, nitrogen doping causes $TiO_2$ on the GO surface to seem uniformly dispersed. X-ray Diffractometer (XRD) spectrums suggest that $TiO_2$ on the GO surface presents an anatase crystal structure; infrared spectrums display the characteristic vibration peaks of the $TiO_2$ and GO. Adsorption performance analysis illustrates that $N-TiO_2/GO$ can provide an adsorption amount of 167.92 mg/g in the same time frame and photocatalysis for visible lights of 57.69%. All data present an excellent linear fitting relationship to the first-order dynamic equation. Therefore, the prepared GO composite materials possess outstanding absorption and photocatalysis performances, providing an experimental basis for sewage treatment and purification using photocatalysis approaches in the future.

**Keywords:** graphene oxide; photocatalysis; sewage treatment; rhodamine B; nanomaterial

## 1. Introduction

Under accelerated social-economic development, living standards are improved significantly; meanwhile, the social urbanization process also gets quickened. Under such a trend, the living environment of humankind has been severely polluted and even destroyed in places, especially water bodies. Wastewater from various industries, including agriculture, chemistry, and medical treatment, damages the ecological environment, arousing increasing concern globally. Statistics suggest that at least 90% of domestic sewage and 1/3 industrial wastewater in China enter the rivers without treatment every year. At least 2/3 of the rivers monitored nationwide are affected by various pollutants [1,2]. These polluted water bodies contain many persistent and refractory biodegradation organic pollutants, such as conjugate dyes, antibiotics, and aromatic phenolic compounds. While these compounds severely impact the human race, they also cause hypoxia in the water bodies and even the death of aquatic organisms [3–5]. Therefore, removing pollutants from water bodies is an arduous task faced by water bodies during purification. Graphene oxide (GO) owns the advantages of no secondary pollution during adsorption, simple operation,

and low costs as a physical adsorbent (GO). Hence, it has become the focus in developing new and efficient adsorption materials.

Countless materials have tested the photocatalysis technique since it was proposed. At present, common photocatalysis agents include $TiO_2$, $Ag_3PO_4$, $Cu_2O$, ZnO, $Bi_2WO_6$, and $Fe_2O_3$. The $TiO_2$ photocatalyst has the advantages of low price, environmental friendliness (it can completely decompose organic pollutants, such as organic dyes, toxic micropollutants, and oil, into small inorganic molecules such as $H_2O$ and $CO_2$), stable photochemical properties, and no secondary pollution, making it widely accepted in wastewater treatment [6]. Many researchers have explored $TiO_2$ photocatalysts. Liu et al. (2019) constructed a new type of low anode bias voltage photo-assisted electrochemical system. Under natural sunlight, the degradation test of bisphenol A was performed for 20 cycles. Results showed that the constructed system had good performance and stability under low bias voltage, without evident oxygen evolution [7]. Walenta et al. (2020) used Auger electron spectroscopy, temperature-programmed desorption/reaction, isotope labeling, and isothermal photoreaction to prepare and catalyze the Pt cluster-loaded $TiO_2$ (110) photocatalyst. They determined that the species on the catalyst surface could be catalytically treated under photocatalytic conditions. Moreover, this method could be extended to various noble metal promoters and other $TiO_2$ modifications [8]. Islam et al. (2020) designed a preparation method for a photocatalytic coating based on nano-$TiO_2$ and acrylate photopolymerization resin. They conducted photocatalytic degradation experiments under sunlight and UV-B light. Results demonstrated that under UV-B light, it took 120 min to degrade about 80% of the 6 ppm methylene blue (MB) solution. In contrast, it took 60 min to degrade about 90% of the same MB solution under sunlight; the mechanism was analyzed as well [9]. Zhang et al. (2021) successfully combined an oxygen or nitrogen-linked heptazine-base polymer (ONLH) with $TiO_2$ nanoparticles to form a $TiO_2$/ONLH nanocomposite that responded to natural sunlight material. Results found that under natural light, $TiO_2$/ONLH could degrade ten drugs. At the same time, $TiO_2$/ONLH could be used to remove drugs in wastewater [10].

GO is a hexagonal lattice nanomaterial with a 2D honeycomb and strong atomic sheets, in which the carbon atoms are arranged according to the sp2 hybrid orbital [11]. Due to its good electrical, optical, and mechanical properties, its application in bioenergy, nano-processing, energy, pharmacology, and environmental governance have become increasingly extensive. Regarding water treatment, GO materials can provide excellent adsorption and filtration performances due to their large specific surface area, porous structure, good electrical conductivity, and outstanding stability. Therefore, GO and its composite materials have become universally accepted in water treatment. At present, water pollutant treatments include the Advanced Oxidation Process (AOP) [12], membrane filtration [13], adsorption [14], coagulation/flocculation [15], and photocatalysis degradation [16]. Photocatalysis degradation is considered an environmentally friendly technique due to the permanent solar energy and non-secondary pollution. Adsorption is inseparable from photocatalysis and is also a prerequisite for photocatalysis. The photocatalytic principle refers to that, under illuminated conditions, electrons will transition from the valence band to the conduction band and form photo-generated electrons before generating photo-generated holes at the valence band. Afterward, they possess redox performance, enabling them to degrade and purify organic pollutants, thereby promoting the conversion and synthesis of substances [17]. Because GO can enhance the catalytic effect of semiconductor materials under the combined action of visible light and ultraviolet light, GO itself has good electrical conductivity and a large specific surface area, which can serve as a carrier to provide attachment points for the photocatalysis materials and quickly transmit photo-generated electrons to avoid accumulation, thereby reducing the possibilities of electron-hole recombination. Hence, the efficiency of the photochemical material catalysis is increased. GO can reduce the forbidden bandwidth of photocatalysis materials. It corrects the disadvantage that photocatalysis materials with larger bandgaps only present photochemical activity in the ultraviolet region and improve visible light uti-

lization, increasing the photocatalysis efficiency. Therefore, GO-based composite materials have a broad application range in water purification.

Given the increasingly scarce water resources, the treatment and purification of pollutants in water bodies are highly urgent and significant for the sustainable development of society and the economy. Guided by the green and sustainable development concept, the innovative points are (1) preparing $TiO_2$ nanoparticles by the sol-gel technique, GO by the modified Hummers technique, and, finally, a new $N-TiO_2/GO$ photocatalysis composite material by hydrothermal synthesis; and (2) using rhodamine B (RhB) as simulated pollutants to analyze the adsorption and photocatalysis properties of the prepared GO and its composite materials. The results are anticipated to provide experimental references for treating and purifying water resources in the future.

## 2. Methodology

The purpose is to use GO materials to treat and purify sewage. GO's chemical structure makes it chemically stable. Its surface does not contain any active groups and is almost inert, making it difficult to interact with other substances. However, its specific surface area is vast, with exceptionally high specific surface energy. According to the second law of thermodynamics, GO will spontaneously undergo irreversible agglomeration through van der Waals forces and other effects, which makes the dispersion of GO in water and organic solvents very poor, thereby greatly limiting its application in sensors, photocatalysis, and water treatment [18,19]. Therefore, the functional modification of GO is significant. The process to prepare GO composite material is presented in Figure 1.

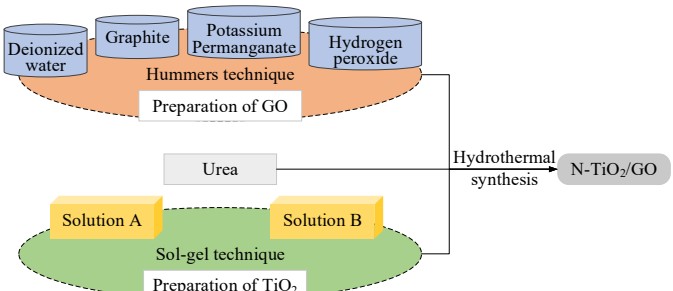

**Figure 1.** The process to prepare GO composite.

First, $TiO_2$ nanoparticles were prepared using the sol-gel technique. Then, GO was manufactured using the modified Hummers technique. Finally, a new $N-TiO_2/GO$ photocatalysis composite material was prepared using hydrothermal synthesis. As a nitrogen source, urea uses non-metal doping to broaden the photoresponse range of $TiO_2$. The photoresponse range of $TiO_2$ is further broadened because GO's excellent performance can suppress the recombination of photo-generated electrons and holes. The huge specific surface area of GO can increase the contact ratio of the photocatalysis materials and pollutants. $TiO_2$, $N-TiO_2$, and $TiO_2/GO$ were prepared as the control groups. The prepared materials were analyzed by Scanning Electron Microscopy (SEM), X-ray Diffractometer (XRD), Fourier-Transformed Infrared Spectroscopy (FTIR), and Raman Spectroscopy and XPS (X-ray Photoelectron Spectroscopy). Moreover, the RhB solution's photocatalysis degradation performance was studied under ultraviolet and visible lights.

### 2.1. GO Preparation

GO is prepared using the modified Hummers technique [20]. First, 23 mL of 98% concentrated sulfuric acid and 0.5 g of sodium nitrate were added to a beaker and evenly mixed using a magnetic stirrer. Second, 1 g of graphite powder was added under an ice-water bath environment. Third, 3 g of potassium permanganate was slowly added after mixing. The mixture was then stirred for 3 h. Next, the beaker was placed into a 35 °C, constant-temperature water bath and stirred for 30 min. Afterward, 46 mL of deionized

water was added slowly. The temperature of the constant-temperature water bath was increased to 98 °C; 30 min later, the mixture was diluted to 200 mL with deionized water and added with 5 mL of 30% hydrogen peroxide. Once the mixture's color turned yellowish-brown, the product was filtered; then, the filter cake was washed with 5% hydrochloric acid solution and deionized water, and finally, separated and centrifuged until the pH value of the supernatant was about 7. To the obtained GO was added deionized water of an appropriate amount, dispersed for 2 h under ultrasonic conditions (the power and frequency were 250 W and 40 kHz, respectively), and freeze-dried for 24 h. The prepared GO was sealed and stored for later use.

### 2.2. TiO$_2$ Preparation

TiO$_2$ nanoparticles were prepared using the sol-gel technique [21]. First, 20 mL of tetrabutyl titanate was added to 80 mL of absolute ethanol, and the mixture was mixed evenly. The resulting solution was marked as solution A. Second, 20 mL of absolute ethanol, 20 mL of deionized water, and 8 mL of acetic acid were mixed to prepare solution B. After the pH value of solution B was adjusted to 2~3 with nitric acid, it was added dropwise to solution A under vigorous stirring. After the titration, a mixed solution was obtained, which was continuously stirred for about 1 h using a magnetic stirrer. Then, the solution was settled for 2 h, forming a yellow transparent gel. The gel obtained was dried in an oven at 100 °C for 12 h, producing yellow crystals. The crystals were ground into a white powder. Finally, the powder was calcined under 500 °C for 2 h to obtain TiO$_2$, with a gradient temperature increase of 5 °C/min. The prepared material was stored for later use.

### 2.3. Preparation of N-TiO$_2$, TiO$_2$/GO, and N-TiO$_2$/GO

The N-TiO$_2$, TiO$_2$/GO, and N-TiO$_2$/GO composite photochemical catalysts were prepared by hydrothermal synthesis [22]. The steps to prepare the N-TiO$_2$/GO material were as follows. First, 0.05 g of GO was dispersed in 80 mL of deionized water; the solution was stirred vigorously for 20 min and ultrasonicated for 1 h. Second, 1 g of TiO$_2$ and 0.1875 g of urea were added to the solution and stirred for 1 h. Third, the suspension was transferred to a 100 mL Teflon-lined stainless-steel hydrothermal reactor and stood at 180 °C for 16 h. Afterward, the suspension was naturally cooled to room temperature. Finally, the suspension was centrifuged at 8000 rpm. After centrifugation, it was washed with deionized water at least three times and vacuum dried at 60 °C for 18 h. The solids were then ground to obtain the N-TiO$_2$/GO composite photocatalysis material.

While preparing the N-TiO$_2$, except for the 0.05 g GO that was not added, the other steps were identical to the N-TiO$_2$/GO preparation.

While preparing TiO$_2$/GO, except for the 0.1875 g urea that was not added, the other steps were identical to the N-TiO$_2$/GO preparation.

### 2.4. Characterization of GO Composite Materials

All samples were characterized in powder form. The surface morphology of the materials was observed under an SEM (JSM–6700F, Japan) and a Transmission Electron Microscope (TEM, Tecnai G2F20 S-TWIN, USA) [23,24]. Before observing the samples under SEM, they were evenly coated on the conductive adhesive glued on the aluminum block, and the target was sprayed with gold using the sputtering coater for 60 s. To observe the samples under TEM, a small amount of the sample was directly dispersed in absolute ethanol. Then, 1 to 2 drops of the sample suspension were dropped onto a clean copper grid. After the solvent evaporates, the sample was loaded, and the appearance of each sample was characterized and observed.

The Cu K$\alpha$ ray is the X-ray radiation source ($\lambda$ = 1.541A0). The XRD patterns in the materials were recorded using a German Bruker AXSD8 XRD [25] at 40 kV and 250 mA, with a scanning step of 0.02°/0.15 s and a scanning range of 2θ = 5°~80°, to characterize the structure of each sample.

The FTIR [26] of the samples was tested using a Nicolet 5700 spectrometer through dry KBr (the sample/KBr ratio is 1:100) at room temperature; the aperture diameter was 4 mm, the spectral resolution was 4 cm$^{-1}$, and the number of scans was 96 times.

An InVia confocal Raman spectrometer [27] was applied to analyze the evolution of the number of GO layers calcined at high temperatures. In the present work, the incident laser with a wavelength of 488 nm was selected. The output power of the laser was set to 30 mW, and the Raman spectrum of the GO was measured after the instrument was adjusted.

The surface of the N-TiO$_2$/GO photocatalyst was analyzed by XPS, and the existing form and chemical valence of the C element in graphene were analyzed. The X-ray source was MGK $\alpha$, the test step was less than or equal to 1 eV, and the sample was about 0.1 g when XPS was used for testing. The binding energy was corrected with C ls as the benchmark, and then the peak fitting was performed.

### 2.5. Adsorption Experiment

RhB is a common dye pollutant in water bodies, a water-soluble, non-biological degradation dye with carcinogenic and mutagenic toxicity [28]. To test the adsorption performance of the prepared composite materials, the solution containing RhB was used as the simulated pollutant. The adsorption experiment was done under neutral pH conditions. Reactions occurred in a 250 mL beaker containing a 20 mg powder sample and 100 mL RhB solution under a dark environment. During the adsorption process, the suspension's uniformity was maintained by magnetic stirring, and a 1 mL sample was taken out for analysis at a set time point. Adsorption dynamics were investigated in the RhB solution at a concentration of 46.9 mg/L. The equation to calculate the adsorption quantity Q (mg/g) of the adsorbent at a specific time t is as follows:

$$Q = (C_0 - C_e)V/M \tag{1}$$

where C$_0$ (mg/L) and C$_e$ (mg/L) refer to the initial concentration of the pollutants in the solution and the remaining concentration of the solution in adsorption equilibrium, respectively; V (L) refers to the volume of the solution; and M (g) refers to the mass of the adsorbent.

### 2.6. Photocatalysis Degradation Experiment

The RhB solution serves as the simulated pollutant in the photocatalysis experiment to assess the materials' photocatalysis performance. Visible light (>420 nm) catalyze the RhB degradation under the irradiation of a 300 W xenon lamp (CEL–HXF300, China Education AU-Light). The experimental voltage is 14 V, the current is 16 A, and the xenon lamp's light intensity is 300 mW/cm$^2$. Then, 20 mg of photocatalysis agent is added to 100 mL of a 46.9 mg/L RhB solution, and magnetically stirred in the dark for 60 min to reach adsorption–desorption equilibrium. Afterward, the mixed solution was illuminated; at the set time point, a small amount of solution was pipetted and filtered with a 0.22 um Poly Tetra Fluoroethylene (PTFE) filter head. A UV-vis spectrophotometer detected the concentration of the filtered RhB solution with a wavelength of 554 nm. The concentration of the RhB solution after reaching the adsorbent equilibrium was taken as the initial concentration (C$_0$). All samples were collected after illumination exposure and washed three times with water to remove residual impurities, and finally, the sample was dried at 80 °C for recycling.

## 3. Results and Discussions

### 3.1. Characterization Analysis

After the samples were prepared, their morphology and structure were analyzed, as shown in Figures 2–5.

The surface morphology of the prepared samples was observed under SEM, and the results are presented in Figure 2. As shown in Figure 2A, GO presents a layered structure, with folds and irregular curvatures on the surface. Figure 2B shows TiO$_2$; the particle size

is not large, the dispersion of the nanoparticles is inconsistent, and some regions have poor dispersion and agglomeration, which is related to the difference in the surface structure of the nanoparticles. Figure 2C shows nitrogen-doped $TiO_2$ with a small particle size and similar morphology to $TiO_2$. Figure 2D,E display $TiO_2/GO$ and $N-TiO_2/GO$, respectively. GO is reduced in the hydrothermal reaction, and $TiO_2$ is distributed on the GO surface. The two materials show similar morphologies; however, $TiO_2$ on the GO surface seems uniformly dispersed in the 2D image due to nitrogen doping.

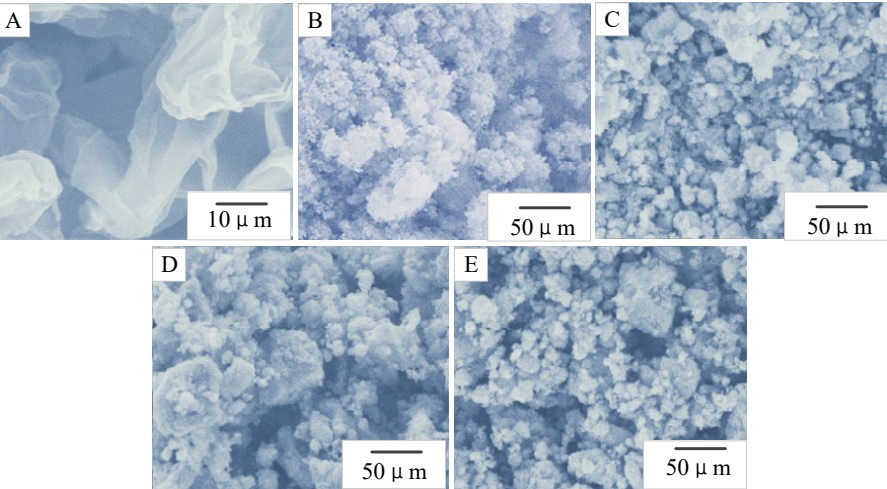

**Figure 2.** SEM images of the GO composite photocatalysis materials ((**A**) GO; (**B**) $TiO_2$; (**C**) $N-TiO_2$; (**D**) $TiO_2/GO$; (**E**) $N-TiO_2/GO$).

The crystal face characteristics of each sample were analyzed using XRD, and the results are displayed in Figure 3. Figure 3b shows the XRD spectrogram of the GO. A strong diffraction peak appears near $2\theta = 10.44°$, which corresponds to the characteristic diffraction peak of the GO (001) plane. The XRD patterns of the $TiO_2$, $N-TiO_2$, $TiO_2/GO$, and $N-TiO_2/GO$ samples all show an anatase crystal structure. When $2\theta$ takes $25.77°$, $37.98°$, $48.11°$, $53.73°$, $55.27°$, $62.75°$, and $75.14°$, respectively, the characteristic diffraction peaks of anatase appear, corresponding to the 101, 004, 200, 105, 211, 204, and 215 crystal planes of anatase, respectively. The characteristic diffraction peak of rutile does not appear, indicating that the hydrothermal synthesis of nitrogen and GO doping will not affect the crystal structure of $TiO_2$ and does not change or destroy the crystal structure of the $TiO_2$. GO's characteristic diffraction peaks do not appear in these four types of samples, possibly due to its reduction during hydrothermal synthesis of the composite photocatalysis agent, forming reduced GO [29]. The characteristic diffraction peak of $TiO_2$ is 25.77, covering the characteristic diffraction peaks of RGO.

An infrared spectrum test aims to observe the characteristic functional groups of the material. Our test results are presented in Figure 4. Figure 4a shows GO's spectrum. The broad absorption peak at $3204\ cm^{-1}$ in the spectrogram is the vibration stretching peak of -OH in the GO structure. The peak at $1606\ cm^{-1}$ is the vibration stretching peak of the C=C skeleton in the GO structure. The peak at $1379\ cm^{-1}$ is C–O's vibration stretching peak of C–OH in the GO structure. The peak at $1081\ cm^{-1}$ is C–O's vibration stretching peak in the C–O–C epoxy group. The peak at $852\ cm^{-1}$ is the vibration absorption peak of the epoxy group. The absorption peaks below $1000\ cm^{-1}$ in Figure 4b–e are the characteristic vibration peaks of Ti–O–Ti of $TiO_2$. Compared with GO's spectrum, the vibration peak intensity of the oxygen-containing functional groups of $TiO_2/GO$ and $N-TiO_2/GO$ is weakened, the characteristic peak of GO disappears, and the spectrum becomes smooth, indicating that GO has been reduced during the hydrothermal process. Simultaneously, some oxygen-containing functional groups and $TiO_2$ oxygen-containing functional groups work together to form Ti–O–C, which coexists with Ti–O–Ti.

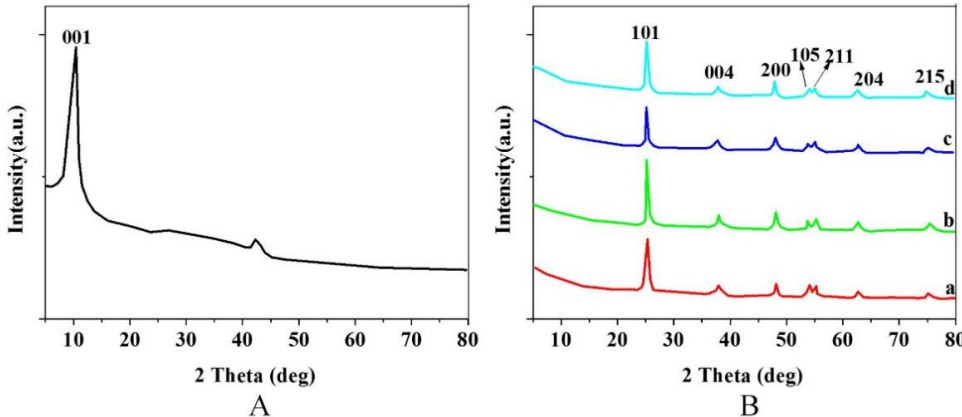

**Figure 3.** XRD spectrograms of the different samples ((**A**) GO; (**B**): a. $TiO_2$; b. GO; c. N-$TiO_2$; d. $TiO_2$/GO; e. N-$TiO_2$/GO)).

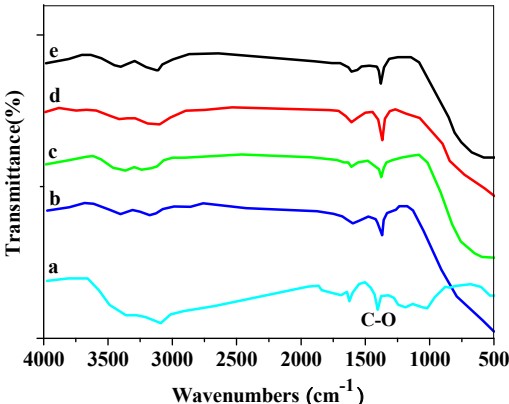

**Figure 4.** FTIR spectrograms of the different samples (a. GO; b. $TiO_2$; c. N-$TiO_2$; d. $TiO_2$/GO; e. N-$TiO_2$/GO).

Figure 5 presents the growth of the graphene prepared in this study, as analyzed by Raman spectrometry. The ratio of the GO's D peak intensity to its G peak (ID/IG) intensity is 0.4 and 0.1, respectively. This shows that when graphene is prepared by the hydrothermal method, the GO defects are reduced, and the quality is significantly better. For flat single-layer graphene, the 2D peak is usually a single peak. The 2D peaks of the prepared GO are fitted; that is, the corresponding illustration in Figure 5 is from the fitting results of the 2D peaks. Although the 2D peaks in the Raman spectrum can all be fitted as a single peak, this cannot prove that it is a single-layer GO because, for GO with more surface wrinkles, disordered stacking between its layers will also produce this result.

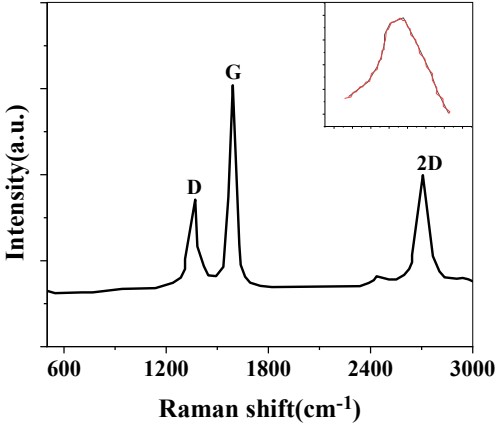

**Figure 5.** Raman spectrum of GO in N-doped GO composites.

Figure 6 is a high-resolution scanning XPS of C ls in the N-TiO$_2$/GO composite. There is a symmetrical Sp2 hybrid C–C/C=C peak at the binding energy of 284.8 eV. In addition, there are oxygen-containing functional groups, such as C–OH, C–O–C, and -COOH in the figure. Compared with the high-resolution XPS scanning of GO in the related literature [30], the diffraction peak intensities of the C–OH, C–O–C, and -COOH of N-TiO2/GO in Figure 6 are significantly lower. It indicates that, in the hydrothermal process, the oxygen-containing functional groups on the surface of the GO are partially reduced in the high-temperature and high-pressure environment inside the reactor, so there may be RGO in the composites. The C=N and C–N bonds in Figure 6 show that nitrogen is doped in graphene. The existence of C–Ti indicates that TiO$_2$ and GO are not simply physical adsorption, but chemical bonding. Therefore, the analysis of C in the composite reveals that the hydrothermal reaction makes the doping of nitrogen and the reduction of GO proceed simultaneously. The combined action of electrostatic adsorption and chemical bond makes the TiO$_2$ and GO closely combined, which improves the interface charge transfer efficiency, effectively inhibiting the recombination of the photoelectrics and holes, increasing the adsorption of pollutants, accelerating the oxidation–reduction reaction, and improving the photocatalytic activity of the composite photocatalyst.

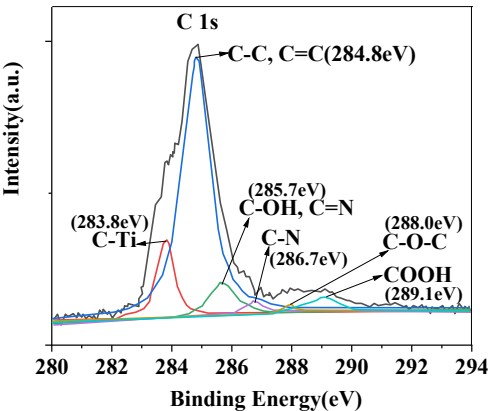

**Figure 6.** High-resolution scanning XPS of C ls in N-TiO$_2$/GO composites.

### 3.2. Adsorption Performance Analysis

The adsorption performance of each sample was analyzed. The adsorption ability and pseudo-second-order adsorption dynamics are illustrated in Figures 7 and 8.

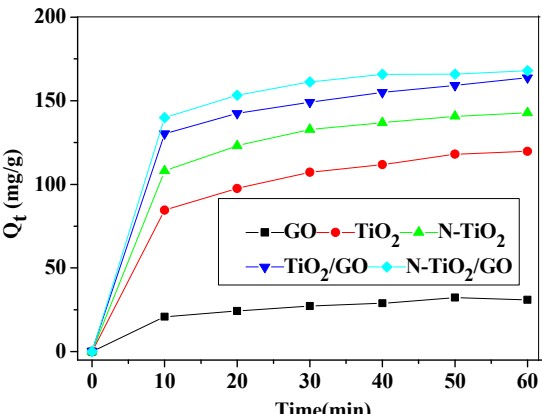

**Figure 7.** RhB adsorption by different samples.

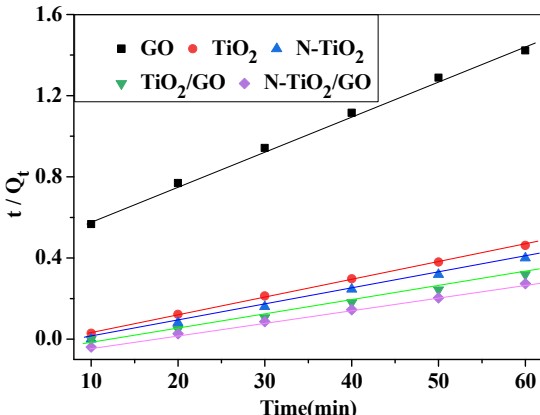

**Figure 8.** Pseudo-second-order dynamics of different samples for RhB adsorption.

Time is an essential factor affecting adsorption performance. As shown in Figure 7, the adsorption rate of the material is very fast, the adsorption amount increases rapidly, and all the adsorption curves show three different stages. The first stage is within 0–10 min, during which the adsorption rate is the fastest, and the amount of adsorption increases linearly with time. The second stage is within 10–40 min, during which the adsorption rate gradually decreases, and the adsorption amount slowly increases until it no longer changes. The third stage is within 40–60 min, during which the amount of adsorption does not change, indicating the adsorption equilibrium. To make an intuitive comparison of the different materials' adsorption rates, the above experimental data were subjected to dynamics simulation, and the results are displayed in Figure 8. All the adsorption processes are in line with the pseudo-two-order adsorption dynamic model. The N-TiO$_2$/GO material has the fastest adsorption rate and the highest adsorption capacity due to the nitrogen doping of the N-TiO$_2$/GO composite material. The TiO$_2$ on the GO surface is uniformly dispersed, providing a high specific surface area and pore volume [31]. Moreover, the interaction between the hydroxyl groups on the nanocomposite surface and the RhB molecule synergistically enhances its adsorption ability.

### 3.3. Photocatalysis Performance Analysis

The photocatalysis performance and mechanism of each sample were further analyzed. The photocatalysis performance of each sample was compared with the degradation effect of RhB under visible light (>420 nm), as shown in Figures 9–11.

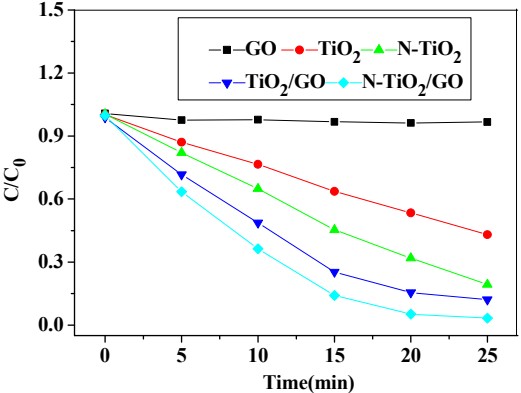

**Figure 9.** Relationships between the RhB photodegradation rates and illumination time of the different samples.

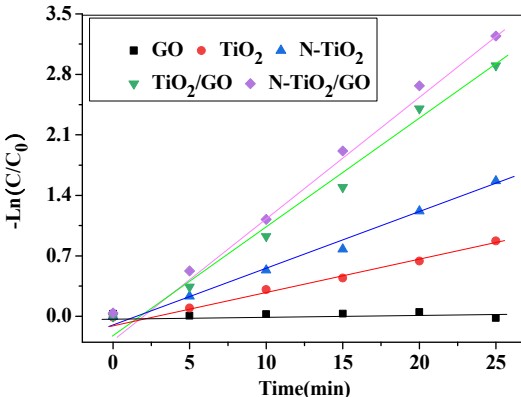

**Figure 10.** Dynamic simulations of the different samples for RhB photocatalysis.

Before illumination, the solution containing the photocatalysis agent and RhB was stirred for 60 min in complete darkness to reach adsorption and desorption equilibrium. The photocatalysis efficiency was calculated through $C/C_0$, where $C_0$ and $C$ are the concentration of RhB at illumination times 0 and $t$ (min). As shown in Figure 9, under the GO photocatalysis agent, after 25 min of illumination, the degradation rate of RhB is only 3.17%. In contrast, the degradation rates of samples containing $TiO_2$ and $N-TiO_2$ are 2.26% and 57.69%, respectively, while those containing $N-TiO_2/GO$ provide a higher RhB degradation efficiency than $TiO_2$ and $N-TiO_2$, indicating that $N-TiO_2/GO$ produce a synergistic degradation effect after chemical recombination. The result of fitting the photocatalysis dynamic equation to the experimental data is presented in Figure 10. All the data share a good linear fitting relationship to the first-order dynamic equation, and the linear correlation coefficient, $R^2$, is greater than 0.972.

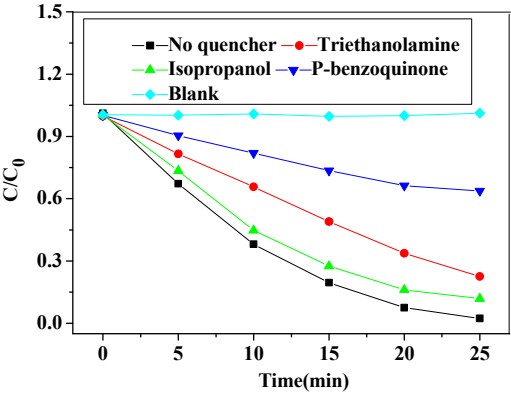

**Figure 11.** Results of free radical capture experiment during RhB degradation.

To explore the mechanism of photocatalysis activity enhancement and RhB degradation, free radicals generated during $N-TiO_2/GO$ photocatalysis were identified by free radical capture experiments. The results are shown in Figure 11. The concentration of RhB does not change without a reagent. Triethanolamine, isopropanol, and p-benzoquinone are used as traps for the holes, hydroxyl radicals, and peroxyl radicals, respectively. The samples were added to isopropanol and illuminated for 25 min; afterward, the photocatalysis degradation efficiency of RhB by $TiO_2/GO$ was slightly reduced from 25% to 94.39%, indicating that the hydroxyl radicals are not the main active species. With the addition of triethanolamine and p-benzoquinone, the degradation rate of RhB drops significantly to 76.37% and 37.35% after illumination exposure for 25 min, indicating that the holes and peroxyl free radicals are the main active substances that attack the RhB molecules, resulting in RhB degradation.

## 4. Conclusions

In the present study, three materials were prepared to treat and purify water resources. First, $TiO_2$ nanoparticles were prepared using the sol-gel technique; second, GO was prepared using the modified Hummers technique; finally, the new N-$TiO_2$/GO photocatalysis composite material was prepared by hydrothermal synthesis. Furthermore, these materials' adsorption and photocatalysis performances were analyzed. Appearance characterization suggests that $TiO_2$ is distributed on the GO surface in the prepared N-$TiO_2$/GO composite material; moreover, $TiO_2$ covers a larger area than the GO surface. Hence, N-$TiO_2$/GO is an excellent adsorbent for water pollutants. It provides the fastest adsorption speed, the highest adsorption capacity, and outstanding photocatalysis performance, providing experimental evidence for nano-composite material preparation and photocatalysis sewage treatment. Still, there are some weaknesses. Only the absorption performance of N-$TiO_2$/GO for RhB, a common dye pollutant, was analyzed in the experiment. However, water pollutants are diversified in real-life settings. Hence, in the future, more research objects should be included in such experiments to provide more reliable references for improving a nanomaterial's adsorption performance.

**Author Contributions:** S.L. and Y.L. thought about the experiment, Z.-L.S. wrote the paper, S.L. and L.-Y.Y. carried out the experiment, and Z.-L.C. and Z.-Z.L. made suggestions for modification. All authors have read and agreed to the published version of the manuscript.

**Funding:** This research received no external funding or This research was funded by [Innovation and Entrepreneurship Plan for College Students in Jiangxi Province, China] grant number [S202010403023].

**Institutional Review Board Statement:** Not applicable.

**Informed Consent Statement:** Not applicable.

**Data Availability Statement:** Data available on request due to restrictions e.g., privacy or ethical.

**Acknowledgments:** Financial support for this project was provided by the Innovation and Entrepreneurship Plan for College Students in Jiangxi Province, China (Grant No.S202010403023).

**Conflicts of Interest:** No conflict of interest.

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
