# Peer review of "Mechanism and Purification Effect of Photocatalytic Wastewater Treatment Using Graphene Oxide-Doped Titanium Dioxide Composite Nanomaterials"

_water, doi:10.3390/w13141915_

Round 1

Reviewer 1 Report

The authors report on the mechanism and purification effects of graphene formation on graphene oxide on sewage in the submitted manuscript. In general, graphene-related nanomaterials' industrial utilization is hampered by scalability issues and lack of uniformity in nanomaterial production. Graphene films are known for their catalytic effects due to the increased surface area on offer. Combining graphene's conduction and catalytic behavior with widely usable industrial applications shall significantly push forward the current status quo.

The topic studied, the novel synthesis procedure discussed, and the quality of the experiments are good, however, it lacks in-depth analysis for the represented area of scalable graphene growth. My main concerns regarding the manuscript are mentioned below:

  1. There are only a handful of techniques that can distinctly distinguish between the varied carbon nanomaterials. Please include the Raman spectrum of as-fabricated graphene oxide, to distinguish it from graphene, graphite, and graphite oxide. Then, comment on the extrapolated thickness values of the GO layers based on the Raman spectra, and the O content, based on the ID/IG ratio obtained. {Ref. ACS applied materials & interfaces 11 (27), 24318-24330 and Scientific Reports volume 6, Article number: 19491 (2016) }To correlate the data further, please provide XRD data to highlight the GO quality thus formed.
  2. Please provide an XRD diffraction pattern on the semi-log scale to establish the absence of impurities in the graphene fabrication process.
  3. The authors also need to calculate the graphene oxide cluster size and layer thickness evolution during graphene formation by employing the ID /IG.
  4. Please comment on how the dynamical simulations are performed for figure 8.

In summary, the manuscript provides exciting data on graphene oxide nanostructures fabrication and their potential application in sewage treatment. The investigations are thorough, and the methodology is correct. The interpretation of the results is mostly convincing but needs further work to convincingly elicit scalable graphene fabrication. These concerns need to be addressed before publication in an archival journal like MDPI Water.

Author Response

Dear Reviewers,

Thanks very much for taking your time to review this manuscript. I really appreciate all your comments and suggestions! Please find my itemized responses in below and my revisions/corrections in the re-submitted files. 

  1. The Raman spectrum of GO has been added. The ID/IG ratio obtained by GO based on the Raman spectrum has been reviewed. The XRD diagram reflects the diffraction peaks excited by each material. The specific values have been explained in the upper paragraph of Figure 4; thus, it is unnecessary to repeat the data in Figure 4 again. Please check.
  2.  By checking the paper, the XRD patterns of TiO2, GO, N-TiO2, TiO2/GO, and N-TiO2/GO have been provided to characterize and analyze the prepared materials. The XRD diffraction peaks generated in each nanomaterial have been analyzed. Specifically, after analysis, the currently displayed XRD image is sufficient. The presence of impurities can be analyzed through the characteristic diffraction peaks that appear so that it is unnecessary to provide an XRD diffraction pattern on a semi-logarithmic scale. In the following work, further analysis will be conducted based on the properties of the prepared materials, and a more detailed understanding of their composition will be given.
  3. Raman spectroscopy has been added to the manuscript, and ID/IG has been used to calculate the growth and changes of GO.
  4. Figure 9 (original Figure 8) shows the photocatalytic kinetics fitting curve, not the result of dynamic simulation. The method of obtaining specific values at different time points in this figure is explained in Section 2.6. At the same time, the possible reasons for the results are explained in detail in the results and discussion section.

Thanks again!

Reviewer 2 Report

Authors synthesized N-doped TiO2/GO photochemical catalyst composite material by hydrothermal method. In addition, the same sample was further used for RhB pollutant removal by adsorption and photochemical catalysis. The manuscript is interesting but needs major revision.

1) Introduction part need to be revised with some recent references of TiO2 based photocatalyst.

2) Author should correct all grammatical and syntax errors.

3) The XRD of GO should be separately presented.

4) Author should carefully check the captions of all figures and correct if needed.

5) Author must compare present results with literature.

Author Response

Dear reviewer,

Thank you very much for taking the time to review this manuscript. I am very grateful for all your comments and suggestions! Please find below my itemized responses and the revisions/corrections in my resubmitted documents.

1.The introduction has reviewed the latest work on TiO 2 photocatalysis.

2.All grammatical and grammatical errors have been corrected.

3.The XRD of GO has been presented separately.

4.The titles of all numbers have been checked.

5.The research results are analyzed and compared with scholars in related fields.

Thanks again!

Reviewer 3 Report

The manuscript describes the Photochemical Catalysis of TiO2/GO dispersions.

1 ) Author has no demonstrate the presence of GO, by the characterization techniques used. It is well known that GO is usually reduce to rGO by hydrothermal conditions (also commented by author). In X ray diffraction there is an absence of (001) peak of GO and no evidence of (002) due to the hydrothermal reduction of GO to rGO. For that RAMAN spectroscopy is needed to observe the presence of GO or rGO in the composite material. Moreover, it is also suggested to collect the spectra from 100 to 3500 cm-1 to observe the TiO2 and GO/rGO bands and compare the spectra of the composites with TiO2; n-TiO2 and GO.

2) It is well known in the literature the influence of sonication in exfoliation and also in the decrease of the lateral size due to ultrasounds. (see for example   ). Author has used large ultrasonication times. Author should describe the ultrasonication process, if it is used bath or tip, power, frequencies, prove, energy by ml, etc

3) Particle size of TiO2 needs to be measured

4) Several typing & language errors can be find in all of the document: most of the subscripts are missing.

5) Title should be reconsidered; mechanism is not fully described (remove mechanism) and also need to be reconsidered based on the GO/rGO and TiO2 needs to be included

6) Replace “possibly due to the destruction of GO’s layered structure”  by “possibly due to the reduction of GO”; It should be demonstrated as commented before

the manuscript indeed discusses work with high potential impact in may be considered for publication after mayor revision.   

Author Response

Dear Reviewers,

Thanks very much for taking your time to review this manuscript. I really appreciate all your comments and suggestions! Please find my itemized responses in below and my revisions/corrections in the re-submitted files. 

  1. The reason why the (001) GO peak and (002) RGO peak did not appear in the XRD of TiO2/GO and N-TiO2/GO in Section 3.1 has been explained in this revision. In the FTIR spectrum, after analysis, the measurement range of the instrument is 500 to 4000 cm-1, which is appropriate to observe the bands of TiO2, GO/rGO, and composite materials. The spectrum of 100 to 500 cm-1 has no obvious stability, and its meaning is not significant.
  2. The power and frequency in the ultrasound conditions have been explained in the method section, which makes the performance of the ultrasound conditions in this paper clearer.
  3. The morphology and particle size of TiO2 have been added to the morphology characterization.
  4. The subscripts have been corrected.
  5. The paper title has been revised, involving the mechanism of sewage treatment (taking Rhodamine B as a pollutant, and analyzing the capture experiment of free radicals during its degradation process, as shown in Figure 9). In addition, TiO2/GO materials have been added.
  6. This issue has been addressed.

Thanks again!

Round 2

Reviewer 1 Report

The authors have made appropriate corrections that make the manuscript suitable for further publication. 

Author Response

Thanks for the valuable comments. 

Reviewer 3 Report

Most of the questions are well addressed, however, the nature of the GO or rGO needs to be determine due to the well known difference in the photocatalytic behaviour of GO/TiO2; rGO/TiO2 at different oxygen content, due to the changes in bad gad in GO/rGO systems. In previous revision was suggested to characterize by raman as an example. If Raman is not available, other techniques such as XPS and characterize C 1s groups, can prove the nature of the GO or rGO in the composite

However, author has included Figure 5 (the growth of GO (1200°C)), that has no from this paper, that the process is hydrothermal at 180ºC. It should be removed.

Author Response

Thanks for the valuable comments. By rechecking the manuscript, the Raman spectra in the manuscript were re-modified to make it consistent with the hydrothermal method of this research; secondly, the properties of GO or rGO were characterized by the characterization C 1s group of XPS analysis,  making the materials prepared in this study and related research more credible. 

Round 3

Reviewer 3 Report

Author has considered all of the question, however, there is a mistake in the C1s interpretation in figure binding energy axe (see for example https://doi.org/10.1016/j.carbon.2018.11.012 and https://doi.org/10.1016/j.carbon.2008.09.045)

Probably, and based on figure position of the peaks, the Ti-C is C-C sp2 and C-C/C=C is C-OH, in agreement with GO or RGO with high oxygen content (low reduction)

Reproduced from https://doi.org/10.1016/j.carbon.2018.11.012: Peak positions of the non-equivalent carbon species, based on literature data, were: aromatic carbon (C-C sp,2 284.4 eV), aliphatic carbon (C-C sp3, 285.0 eV), hydroxyl (C-OH, 285.7 eV), epoxy (C-O-C, 286.7 eV), carbonyl (C=O, 288.0 eV) and carboxyl (O-C=O, 289.1 eV)

Reproduced from https://doi.org/10.1016/j.carbon.2018.11.012: Peak positions of the non-equivalent carbon species, based on literature data, were: aromatic carbon (C-C sp,2 284.4 eV), aliphatic carbon (C-C sp3, 285.0 eV), hydroxyl (C-OH, 285.7 eV), epoxy (C-O-C, 286.7 eV), carbonyl (C=O, 288.0 eV) and carboxyl (O-C=O, 289.1 eV)

Author Response

Thank you for your valuable comments.  By re-examining the manuscript, the interpretation of C1s in the graph binding energy ax in the manuscript was revised to make this research more credible.
